https://doi.org/10.1038/s42003-021-02173-3　　**OPEN**
# Comparative analysis of mite genomes reveals positive selection for diet adaptation

Qiong Liu [1], Yuhua Deng[2], An Song[3], Yifan Xiang [1], De Chen [4✉] & Lai Wei [1✉]

Diet is a powerful evolutionary force for species adaptation and diversification. Acari is one of the most abundant clades of Arachnida, exhibiting diverse dietary types, while the underlying genetic adaptive mechanisms are not fully understood. Based on comparative analyses of 15 Acari genomes, we found genetic bases for three specialized diets. Herbivores experienced stronger selection pressure than other groups; the olfactory genes and gene families involving metabolizing toxins showed strong adaptive signals. Genes and gene families related to anticoagulation, detoxification, and haemoglobin digestion were found to be under strong selection pressure or significantly expanded in the blood-feeding species. Lipid metabolism genes have a faster evolutionary rate and been subjected to greater selection pressures in fat-feeding species; one positively selected site in the fatty-acid amide hydrolases 2 gene was identified. Our research provides a new perspective for the evolution of Acari and offers potential target loci for novel pesticide development.

[1] State Key Laboratory of Ophthalmology, Zhongshan Ophthalmic Center, Sun Yat-sen University, Guangzhou, China. [2] Clinical Research Institute, The First People's Hospital of Foshan, Foshan, China. [3] ShaanXi JunDa Forensic Medicine Expertise Station, The Fourth Military Medical University, Xi'an, China. [4] MOE Key Laboratory for Biodiversity Science and Ecological Engineering, College of Life Sciences, Beijing Normal University, Beijing, China. ✉email: chende@bnu. edu.cn; weil9@mail.sysu.edu.cn

Diet is one of the most fundamental aspects of an animal's biology and is a powerful evolutionary force for species adaptation and diversification[1–3]. Different diets have been acknowledged to generate various physiological, biochemical, and morphological adaptations[4]. With the development of genome sequencing, the genetic basis of dietary adaptation has been revealed gradually, providing new and valuable insight into evolutionary biology[5–8].

The abundant species diversity of mites and ticks (Arachnida: Acari) has inspired acarologists and ecologists to understand their evolutionary processes and ecological characteristics for decades[9]. Approximately 55,000 species have been described[10], and the total species number is estimated to be much larger[9,10]. It was reported that the diverse lifestyles in Acari (e.g., free-living, symbiotic, and parasitic) serve as an important driver of its species diversity[11]. The dietary lifestyles of Acari range from predatory to parasitic[9]. Compared with spiders with a predatory diet, Acari exhibit several new diets, such as herbivory and parasitism (e.g., blood and fat bodies), as well as some transitional states such as the consumption of skin exudates, decomposing biomass, and fungi[9,12]. These different specialized diets have generated a number of physiological, biochemical, and morphological adaptations[4]. For example, herbivorous mites have developed a long and stout stylet-like structure for feeding on plant juice[13,14]. A key mechanism of phenotypic adaptation is genetic adaptation, including changes in functional genes, metabolic and regulatory pathways, and even amino acids[5,6,15]. However, the genetic mechanisms underlying the dietary adaptation of Acari have not yet been systematically discussed in a comparative manner.

According to the phylogeny of Arachnida[16,17], Acari consists of two major clades: Acariformes and Parasitiformes[18]. In Acariformes, some mites maintained predatory diets, while some obtained transitional, herbivorous (e.g., *Tetranychus urticae)* or blood sucking (e.g., *Leptotrombidium deliense*) diets[19,20]. In Parasitiformes, ticks (Ixodida) became obligate hematophages and some mites also developed parasitic diets, feeding on blood and fat bodies, respectively[12,21]. The distributions of different diets across the phylogeny of Acari provide an opportunity to understand the genetic mechanisms of dietary adaptation and evolution.

In the current study, we performed comparative analyses of genomes from fifteen Acari species in four major orders and with five kinds of diets. We constructed a well-resolved fossil-calibrated phylogenetic tree and revealed the genomic evolution of Acari. Furthermore, we investigated the genetic adaptation associated with three specialized diets, blood sucking, herbivory, and fat feeding, and found important sites and pathways that underwent adaptive convergent evolution. Our findings, with further experimental validation, can be used as potential targets for drug and pesticide research to control herbivorous and parasitic mites.

## Results and discussion

A total of sixteen Arachnida genomes, including those of one tick, fourteen mites, and the velvet spider as an outgroup, were obtained from GenBank (details in "Methods"). The sixteen species displayed all five kinds of diets, including three predacious species, two herbivorous species, three fat-feeding species, three blood-feeding species, and five transitional species (Table 1). Six genomes were reannotated (marked in Table 1) using ab initio prediction and homology prediction methods together (details in "Methods"). To evaluate genome completeness, we examined the genome and genes by Benchmarking Universal Single-Copy Orthologs (BUSCO)[22]. We observed high BUSCO scores of genome completeness (average 90.9%) and gene completeness (average 79.9%) (Supplementary Table 1). This result suggested that the assemblies of the sixteen genomes were of high quality for downstream comparative analyses.

To reveal the genomic signatures of dietary adaptations in Acari, we constructed a genome-wide phylogenetic tree based on 48,831 nucleotides. Based on three fossil calibration points and a relaxed molecular clock, the divergence time between Acari and spiders was estimated to be ~477.6 million years ago (Mya) (Fig. 1). Moreover, the divergence between Sarcoptiformes and Trombidiformes was estimated to be ~321.8 Mya, and the divergence between Ixodida and Mesostigmata occurred ~336 Mya (Fig. 1). The phylogenetic tree and time estimates of key nodes were consistent with those in recent studies[16,23–25].

Relaxation of selective constraints on and loss of function of protein-coding genes may occur during dietary shifts[6,26]. A total of 1,210 pseudogenes were identified across all species (Fig. 2a, Supplementary Data 1). Blood-feeding species with larger genomes had a higher ratio of pseudogenes (Fig. 2a), supporting the idea that pseudogenes act as a determinant of genome size evolution[27]. The functions of these pseudogenes, most of which were involved in signal transduction, transport, catabolism and so on, were similar across species (Fig. 2b). A total of 65, 23, 54 and 27, and 29 pseudogenes were detected to be common in the groups fed the five diets, predation, blood, plant, fat, and others (transitional type), respectively (Fig. 2a, Supplementary Data 2). Among these pseudogenes, the genes for ribonuclease H were pseudogenized in both the blood-sucking group and fat-feeding group (Supplementary Data 2), which may be related to their parasitic lifestyles. Interestingly, no pseudogenes were shared across all dietary groups, implying that different patterns occurred during the dietary shift (Supplementary Data 2). Since the pseudogenes are nonfunctional, subsequent analyses based on functional genes were carried out after removing the pseudogenes.

To detect signals of positive selection in Acari with specialized diets, we performed phylogenetic analysis by maximum likelihood (PAML) branch-site model (model = 2, and NSsites = 2) tests for the single-copy orthologous genes. As a result, 530 positively selected genes (PSGs) were identified, and 291, 56, 73, and 110 PSGs were detected for the plant, predation, blood, and fat diets, respectively (Supplementary Data 2). Kyoto Encyclopedia of Genes and Genomes (KEGG) enrichment analysis showed that the PSGs were partly correlated with the ability to digest different food types, such as Porphyrin and chlorophyll metabolism in the plant dietary group and Arachidonic acid metabolism in the fat dietary group (Fig. 2c, Supplementary Data 2). Gene ontology (GO) enrichment analysis also revealed biological processes correlated with different diets, such as organonitrogen metabolism in the herbivorous group (Supplementary Fig. 1 and Supplementary Data 2). To compare the overall pressure of natural selection on different food preferences, a total of 439 PSGs in four dietary groups (at least two species for each group) were selected to conduct cross-sectional analysis (Supplementary Data 2). Herbivorous mites were subjected to the strongest natural selection, containing 265 of the 439 PSGs (Fig. 2d). The number of PSGs under intense selection (PSGs with more than 20 positively selected sites) was much larger in the blood-sucking group than in the other groups (Fig. 2d), suggesting that blood-sucking mites were subjected to stronger selection pressure on specific genes during dietary shifts. Some of these intensely selected genes are related to blood digestion, such as carboxylesterase and serine/threonine-protein phosphatase (Supplementary Data 2).

Gene family expansion and contraction are regarded as another key source of adaptive function[28,29]. Hence, we conducted a

**Table 1 Genomes information used in this study.**

| Latin name | Common name | GenBank AssemblyAccession | Total size of contigs | Contig N50 | Gene number | Diet | Taxonomy |
|---|---|---|---|---|---|---|---|
| Ixodes scapularis[74] | Black-legged tick | GCF_002892825.2 | 2,081,329,876 | 835,681 | 24,489 | Blood sucking | Parasitiformes |
| Metaseiulus occidentalis[75] | Predatory mite | GCF_000255335.1 | 151,323,873 | 200,706 | 11,603 | Predation | Parasitiformes |
| Dermanyssus gallinae[76]* | Roost mite | GCA_003439945.1 | 959,010,206 | 278,630 | 42,159 | Blood sucking | Parasitiformes |
| Tropilaelaps mercedesae[77] | Honeybee mite | GCA_002081605.1 | 326,213,305 | 13,741 | 15,190 | Fat feeding | Parasitiformes |
| Varroa destructor[78] | Honeybee mite | GCF_002443255.1 | 368,670,960 | 201,886 | 10,241 | Fat feeding | Parasitiformes |
| Varroa jacobsoni[78] | Honeybee mite | GCF_002532875.1 | 365,177,116 | 96,030 | 10,724 | Fat feeding | Parasitiformes |
| Steganacarus magnus[79]* | Oribatid mite | GCA_000988885.1 | 112,750,608 | 1727 | 9990 | Transitional | Acariformes |
| Hypochthonius rufulus[79]* | Soil mite | GCA_000988845.1 | 171,814,378 | 3254 | 6285 | Transitional | Acariformes |
| Sarcoptes scabiei[80] | Scabies mite | GCA_000828355.1 | 56,251,741 | 11,383 | 10,644 | Transitional | Acariformes |
| Dermatophagoides farinae[81]* | House dust mite | GCA_002085665.1 | 91,934,661 | 188,869 | 15,394 | Transitional | Acariformes |
| Psoroptes ovis[82]* | sheep scab mite | GCA_002943765.1 | 63,214,126 | 2,279,290 | 12,041 | Transitional | Acariformes |
| Leptotrombidium deliense[83] | Chigger | GCA_003675905.2 | 117,276,895 | 1422 | 15,096 | Blood sucking | Acariformes |
| Dinothrombium tinctorium[83] | Velvet mites | GCA_003675995.1 | 180,156,552 | 16,116 | 19,258 | Predation | Acariformes |
| Brevipalpus yothersi[84]* | Flat mite | GCA_003956705.1 | 70,567,388 | 56,520 | 8,245 | Herbivory | Acariformes |
| Tetranychus urticae[33] | Spider mite | GCF_000239435.1 | 89,613,205 | 212,780 | 18,414 | Herbivory | Acariformes |
| Stegodyphus mimosarum[85] | Velvet spider | GCA_000611955.2 | 2,694,371,924 | 46,340 | 27,235 | Predation | Araneae |

Star-tagged species indicate that their genomes have been reannotated

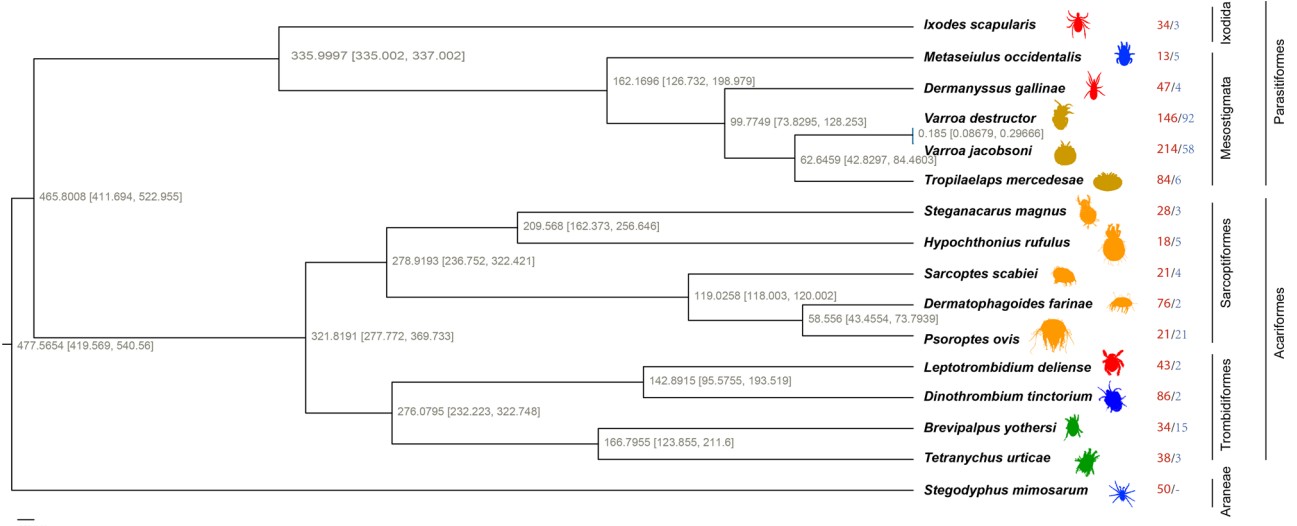

**Fig. 1 The genome-wide phylogenetic tree of Arachnida.** The divergence time was estimated based on a maximum likelihood phylogenomic tree of 16 Arachnida species and three fossil calibrations. All nodes have 100% support according to 1,000 bootstraps. The estimated divergence times are displayed with 95% confidence intervals (in square brackets). Blood-sucking species in red; fat-feeding species in olive; herbivorous species in green; predatory species in blue; and others/transitional in orange. The same colour theme is used in the other figures. The numbers to the right of the species indicate the records of gene family expansions and contractions, with red for expansions and blue for contractions.

protein family (Pfam) domain-based comparative analysis of the sixteen species. Expansion or contraction of a gene family in at least two species in each dietary group was considered a signal of dietary adaptation (details in "Methods"). Fat-feeding mites had more expanded gene families than other dietary groups (Fig. 1); however, no common expanded gene family was found in this group. On the other hand, several gene families related to detoxification and blood coagulation inhibition were expanded in the herbivorous and blooding-feeding mites, respectively (Supplementary Table 2). The detailed genetic basis for each dietary group is presented in each of the following sections.

**Adaption to plant-feeding.** The capability to detoxify plant allelochemicals is considered a key factor driving radiation in herbivorous arthropods[9,30]. In the Acari, Tetranychidae (spider mites, e.g., *T. urticae*) and Tenuipalpidae (false spider mites, e.g., *Brevipalpus yothersi*) are exclusively phytophagous and include major agricultural pests[31]. *T. urticae* is also known for its ability to develop rapid resistance to pesticides[32]. Therefore, we focused on the adaptive traits with a detoxification function in the two herbivorous mites. Consistent with previous reports in several plant-feeding species[33], we observed significant expansion of Cyp18a1, a cytochrome P450 enzyme, at the root of the herbivorous clade (OG0000061, $P < 0.01$) (Fig. 3a, Supplementary Table 2). To confirm this signal, we rescanned all Cyp18a1 genes from our sixteen genomes based on the amino acid sequences of the P450 family and performed manual filtering. The copy number of Cyp18a1 in herbivorous mites, especially in *T. urticae*, was twice as high as that in other mites (Supplementary Data 2). We further collected sixteen RNAseq datasets from four populations of *T. urticae* from GenBank to check whether detoxification-related genes had increased expression (details in "Methods"). As expected, nearly half of the 50 most highly expressed genes were involved in cellular transport and ion binding, which are essential for toxin metabolism[34,35] (Supplementary Fig. 2). Additionally, KEGG pathway enrichment analysis of the 200 most highly expressed genes (Supplementary Data 3) revealed significant enrichment in ATP-binding cassette (ABC) transporters, which are major detoxification families, including the Carboxyl/Cholinesterase (CCE), Cytochrome P450,

and Glutathione-S-Transferase (GST) families[36–38] (Fig. 3b; Supplementary Data 2). The GO item "response to external stimulus", which was enriched in KEGG pathway enrichment analysis, is related to stress responses and handling toxin stimuli (Fig. 3c; Supplementary Data 2). In summary, gene family expansion, as well as gene expression analyses, suggest that herbivorous mites are prominent in their detoxification ability, which may largely contribute to their tolerance to plant allelochemicals and insensitivity to chemical insecticides[39].

Herbivorous mites are known to use odours to select fresh food, locate mates, avoid interspecific competitors and escape predators[40,41]. Among the PSGs found in the herbivorous mites, three genes of the type III heat-shock protein-40 family (Hsp40), namely, *DNAJA2*, *DNAJC13*, and *DNAJC17*, and two genes of Hsp90, namely, *MLH1* and *HSP90A*, displayed the same amino acid changes (Fig. 4). The expression of Hsp25 and Hsp70 was found to increase when exposed to environmental odourants, such as food odour[42,43]. Hsp40 and Hsp90 are often coexpressed with Hsp70, playing a major chaperone role[42,44,45]. Changes in amino acids in Hsp40 and Hsp90 may adjust the expression and/or function of these proteins and make the olfactory response system more conductive. In addition, the expanded P450 superfamily found in the gene family analysis is also involved in the degradation of odour molecules in addition to detoxification[46]. Sensitive olfaction and strong detoxification abilities may provide a strong genetic basis for herbivorous mite feeding on angiosperms.

**Adaptations to blood-sucking.** Blood feeding has evolved several times in Acari (Fig. 1). Adaptive phenotypic convergence has been found in blood-feeding Acari, such as stylet-shaped piercing proboscides and teeth-like mouthparts[47]. Our genomic convergence analysis revealed that nine gene families have undergone significant expansion, including serine protease inhibitor (serpin family member), thromboxane A (*TxA*) synthase, and carboxypeptidase vitellogenic-like (*CPVL*) (Fig. 5; Supplementary Table 2). The serpin family directly inhibits blood coagulation factor XIa or enhances the inhibition of blood coagulation factor Xa via Protein Z[48], which is highly relevant to the blood-sucking process. To quantify the number of genes in the serpin family in each species, we rescanned all serpin family genes from the sixteen genomes

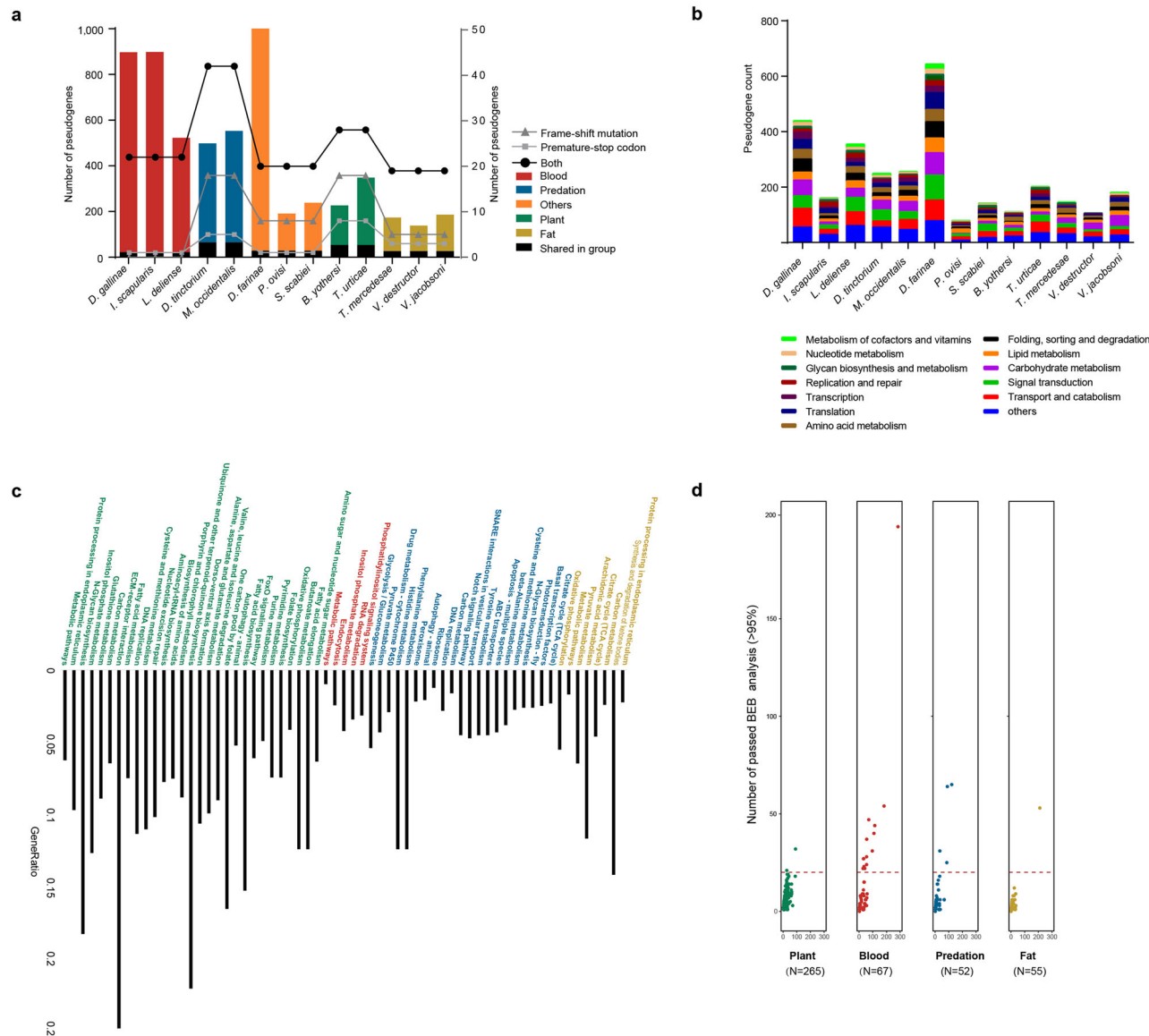

**Fig. 2 Pseudogene and PSG profiles of different dietary groups. a** The overall numbers of pseudogenes detected in each group are shown in different colours reference to left y-axis, and the numbers of shared pseudogenes in each group are shown in black. The pseudogenes in each group are divided into three types, namely, frame-shift mutation, premature-stop codon, and both, and the ratios of types are displayed by lines reference to right y-axis. **b** The functional classification of the pseudogenes in each group. **c** KEGG enrichment analysis of the PSGs. **d** PSGs generated by the orthologous genes annotated in at least two species in each dietary group. The number in parentheses represents the number of PSGs in the group. The dot plot shows the numbers of positively selected sites across four different dietary groups. The x-axis indicates the number of loci detected as being under selection pressure, and the y-axis indicates the number of positively selected loci with a BEB posterior probability greater than 95% in PAML. PSG positively selected gene, KEGG Kyoto Encyclopedia of Genes and Genomes, BEB Bayesian empirical Bayes.

based on the amino acid sequences of the serpin domain and performed manual filtering. As expected, blood-sucking species had more serpin family genes than the other species (Fig. 5; Supplementary Data 2), and expansion was observed for all serpin family genes (Supplementary Fig. 3). *TxA* is an enzyme producing thromboxane A2 (*TXA2*), which can cause vessel constriction and platelet activation and aggregation[49]. In addition, *TxA* can combine cytokines and inflammatory mediators to activate the coagulation cascade[50]. The seemingly antagonistic functions of the serpin family (anticoagulant) and *TxA* (coagulation) may thus provide a fine control system for blood-feeding diets.

Other important features in the blood-sucking process include haemoglobin digestion and detoxification of xenobiotic factors. The expanded *CPVL* gene family is regarded as a member of the serine carboxypeptidases, which are characterized as the main enzymes acting in blood digestion[51–54]. Among the PSGs, *PAQR5* (OG0000879, *P* = 0.02) and *CaMKII* (OG0000260, *P* = 0.05), which are involved in blood pressure management, digestion, and absorption of vascular contents[55–57], were found to be under strong positive selection (Supplementary Data 2). There was one gene, Carboxylesterase (*CES2*), with more than 200 sites under positive selection (Fig. 2d, Supplementary Data 2). *CES2* is recognized as a highly conserved metabolic pathway involved in the metabolism of endogenous and exogenous compounds, which is the key to detoxification during blood feeding[58,59]. The high number of nucleotide sites under selection in *CES2* probably modifies the activity of the coded enzyme, significantly increasing the detoxification abilities in these blood-feeding Acari[60].

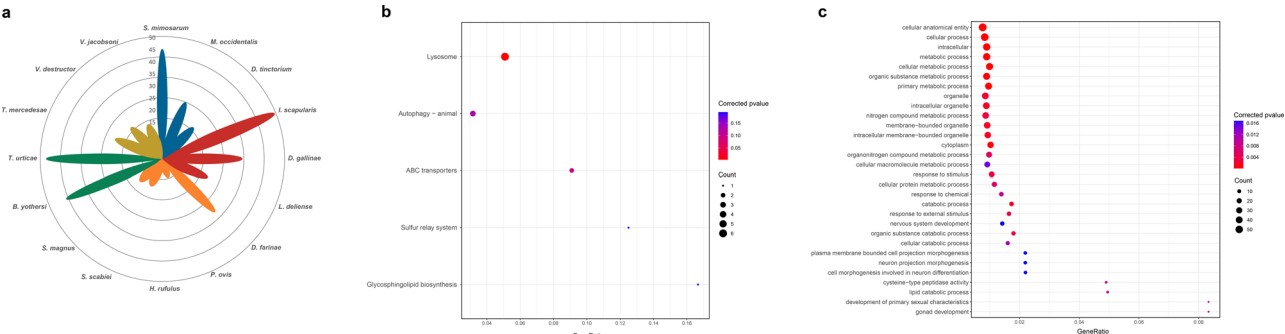

**Fig. 3 Adaptation of herbivorous mites. a** Gene counts related to the P450 gene family of sixteen species. **b** KEGG enrichment of the 200 most highly expressed genes in herbivorous mites. **c** GO enrichment of the 200 most highly expressed genes in herbivorous mites. KEGG Kyoto Encyclopedia of Genes and Genomes, GO Gene Ontology.

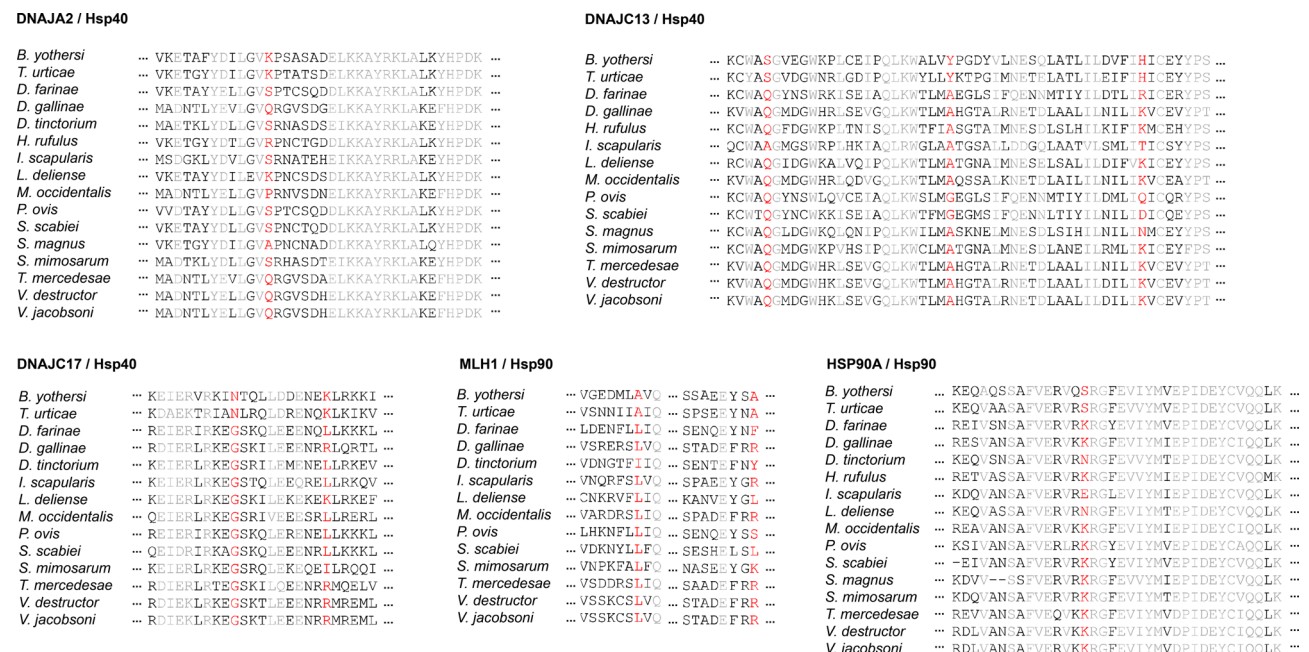

**Fig. 4 The positively selected sites in HSP40 and HSP90 genes.** Red indicates convergence sites.

**Adaption to fat-feeding.** Another major dietary shift in Acari occurs in honeybee mites, which have attracted attention due to their damage to the beekeeping industry worldwide[21]. The identification of the genetic basis underlying fat-feeding habits could contribute to more effective measures for pest prevention, such as the development of new insecticides. KEGG enrichment analysis showed that the 110 PSGs found on the honeybee mite branch were significantly enriched in the Arachidonic acid metabolism pathway (dme00590, P = 0.036) (Fig. 6a, Supplementary Data 2). One positively selected site, W516F/Y, in the fatty-acid amide hydrolases 2 (FAAH2) gene was TTT(F) or TAT (Y) in fat-eating honeybee mites, different from the TGG(W) in other mites (Fig. 6b). The amino acid substitution in FAAH2 occurs in the amidase signature (AS) domain region (Fig. 6b) and may lead to a change in hydrolase activity to influence the ability of FAAH2 to generate free arachidonate acid (AA) from ananadamide. AA is indispensable for the development of the nervous and immune systems, and anandamide is one kind of enriched lipid in honey bees[61]. Compared with the absence of FAAH2 in some mammals such as rats and low-fat-diet mice[62], the change in W516F/Y may enhance the generation of AA from an anandamide-enriched diet. Further experimental validation is needed to confirm this relationship.

To detect the evolutionary pressure on the lipid metabolism system, we retrieved 122 genes related to fat metabolism from the KEGG database and calculated the ratios between the nonsynonymous substitution rates (dN) and synonymous substitution rates (dS) of genes on branches with different diets. Significantly higher dN/dS ratios were observed for the fat-feeding branches (Student's t test, P < 0.01) (Fig. 6c, Supplementary Data 2), which indicates that lipid metabolism genes have evolved at a faster rate and been subjected to greater selection pressures in lipophilic species.

Instead of gene family expansion, we found that three common gene families contracted in fat-feeding mites, including TxA synthase, Serine carboxypeptidase, and Serine protease inhibitor, which were found to be expanded in blood-sucking mites (Supplementary Table 2). This phenomenon suggests a significant difference in dietary adaptation between honeybee mites and blood-feeding mites and provides genetic evidence for the controversy over whether honeybee mites eat blood or lipids[12].

## Conclusion

In the current research, we carried out comparative genomic analyses of sixteen Arachnida species with five different dietary types to explore their genetic evolution and adaptation. We found

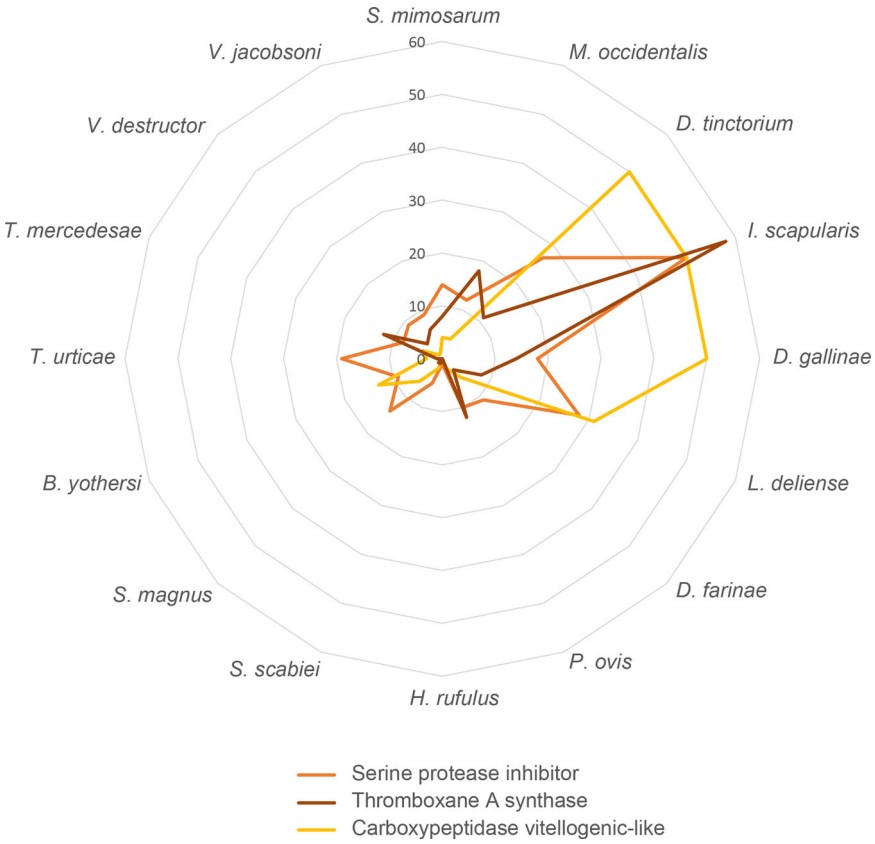

**Fig. 5 Gene family expansion in blood-sucking mites.** Gene counts related to three haematophagous traits of sixteen species.

different genetic bases underlying different diets, mainly related to the need to handle different food types, including increased abilities to find (olfaction), prepare (detoxification) and digest (metabolism) food. Several candidate genes, with further experimental validation, could be used as target loci for novel pesticide development, especially for controlling herbivorous mites and honeybee mites. Future studies on the dietary evolution of the Acari will be important for improving pest control and elucidating host-parasite coevolution.

## Methods

**Genome generation and gene annotation**. All the Acari genomes analyzed in our study are available as public resources from the NCBI (ftp://ftp.ncbi.nlm.nih.gov/). The steps for inclusion were as follows: (1) All Arachnida species in the database (before 2020.06) were searched as candidates; (2) the genome with the best completeness score was selected as the representative if there were two or more genomes for one species; (3) species with genome completeness less than 80% were eliminated; (4) gene prediction was conducted for genomes lacking gene annotation information; and (5) contigs less than 1 kb in length were excluded from the whole analysis. The repetitive elements in the genome sequence were identified by RepeatMasker[63] (version 4.0.7) with Repbase (version 16.02). For gene prediction, the BRAKER pipeline[64] (version 1.9) was used, and the protein sequences of the velvet spider, scabies mite, honeybee mite, chigger, velvet mite, spider mite, predatory mite, dust mite, and tick genome were applied as references for homology-based gene prediction. A link to the annotated results is provided in the Data Availability section.

**Pseudogene detection**. To reduce the heterogeneity of background data, we selected the three genomes with the best quality (those of *D. farina*, *P. ovis*, and *S. scabiei*) in the transitional group as representatives. GeneWise (version 2.4.1) was used for gene alignment, while the protein-coding sequences of a spider (*Stegodyphus mimosarum*) were used as references. The minimal sequence identity was set to 80%, and the cut-off E-value was set to $10^{-5}$. The genes with frame-shift mutations (SNPs or indels) or premature-stop codons were annotated as pseudogenes.

**Phylogenetic tree construction**. Protein sequences with a short length (<50 amino acids) or premature-stop codons were excluded. Orthologous genes were inferred by OrthoFinder (version 2.3.8) with DIAMOND software (version 0.9.24). An expectation value was set to $10^{-5}$. Single-copy orthologous genes were generated from the OrthoFinder result, and conserved blocks of each gene were extracted by Gblocks (version 0.91b). Then, we combined all the conserved regions of every orthologous gene. A total of 50,502 sites were concatenated to build one supergene sequence for each species, which was adopted to construct the phylogenetic tree. The supergene phylogeny was constructed with RAxML[65] (version 8.2.12) under the GTRGAMMA model. A total of 200 bootstrap replicates were conducted, and the tree with the highest likelihood score was picked as a standard tree. Based on the topology of the standard tree, the branch length of each gene was estimated with its sequence. Then, genes with extremely biased branch lengths were excluded. After removing all biased genes, the species tree was reconstructed based on 48,831 conserved sites by RAxML with 1,000 bootstrap replicates following the command raxmlHPC-PTHREADS-SSE3 -x 12345 -p 12345 -# 1000 -m GTRGAMMA.

**Divergence time estimation**. MCMCTREE in PAML[66] (version 4.9i) was used to calibrate divergence time via the Markov chain Monte Carlo method. Fossil time (*Steganacarus magnus* versus *Hypochthonius rufulus*: 394 - 571 Mya; *Dermatophagoides pteronyssinus* versus *Sarcoptes scabiei*: 119 Mya; and *Ixodes scapularis* versus *Dermanyssus gallinae*: 336 Mya) was determined from the Time Tree database (http://www.timetree.org). A relaxed-clock model (clock = 2) was established, and MCMC was performed (burnin = 4,000,000; sampfreq = 100; nsample = 100,000). Other parameters were set to the default, and the Baseml program was run in duplicate to check for convergence.

**Gene family analysis**. The protein sequences of sixteen Arachnida genomes were generated from the annotated results. We removed the proteins with premature-stop codons or a length less than 50 amino acids. The remaining proteins were used to perform protein clustering by OrthoFinder with the all-to-all DIAMOND method and an E-value = $10^{-5}$. A dataset of 39,474 clusters was generated, and some of the clustered were removed due to a low annotation ratio of less than 50 percent of all species. Then 5,718 remaining gene families were subjected to expansion and contraction analysis using CAFÉ[67] (version 4.2.1), and a stochastic birth-and-death model was adopted. The phylogenetic tree with divergence time was obtained with MCMCTREE analysis in PAML.

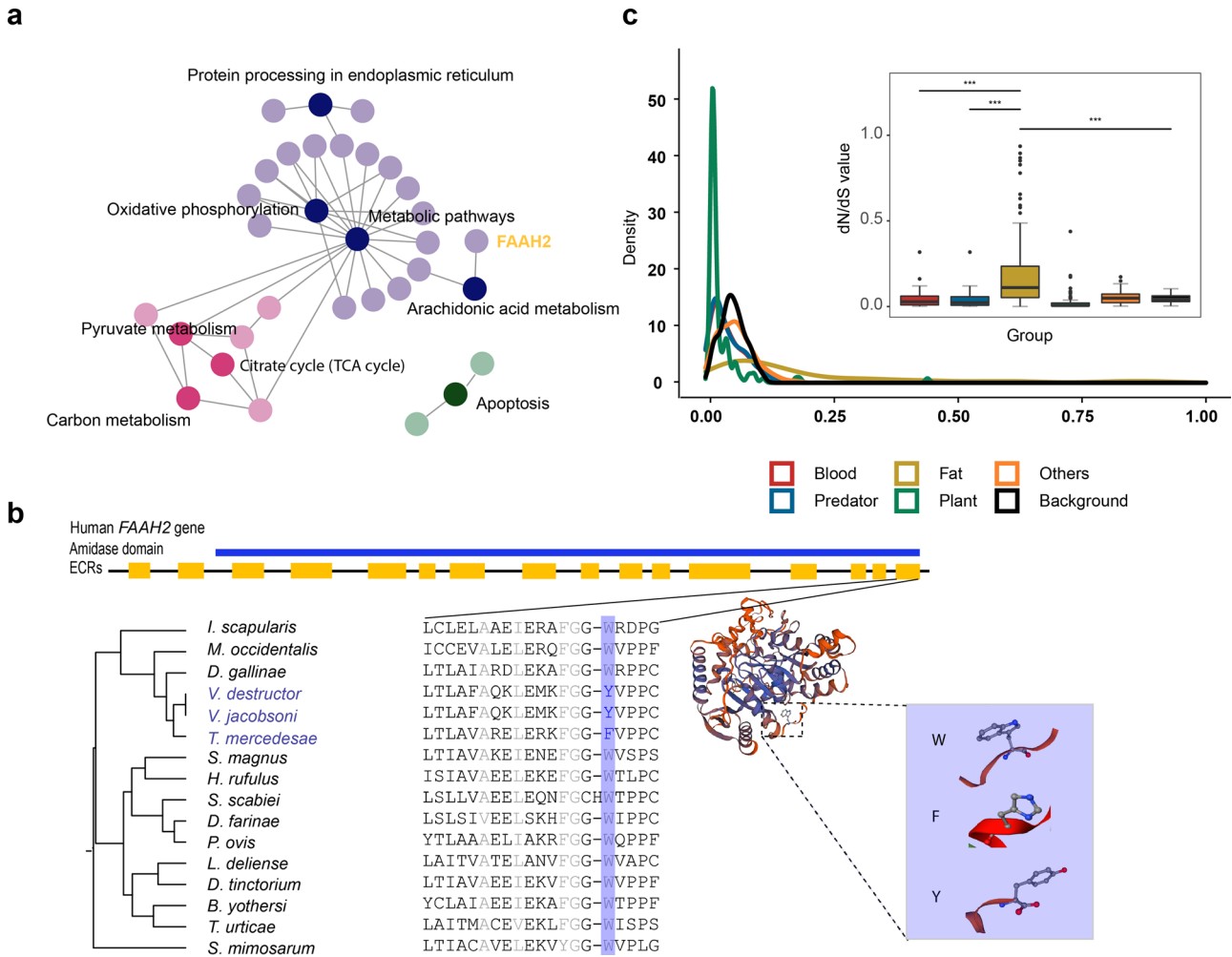

**Fig. 6 PSG and dN/dS values of the lipid metabolism system. a** KEGG pathway enrichment of the PSGs in the fat-feeding dietary group reveals that *FAAH2* is involved in arachidonic acid metabolism. **b** Structural domains of the *FAAH2* protein and comparison of the positively selected amino acid substitutions among the genomes. The site is W516F/Y, and all groups except the fat-feeding group show strict conservation of W amino acids. The structural domain annotation was derived from the NCBI database. **c** Density of dN/dS values of the lipid metabolism gene set in five dietary groups and the background. A total of 122 genes related to the development of lipid metabolism were retrieved from the KEGG, and we calculated the dN/dS ratios on the branches connected to fat-feeding, blood-feeding, herbivorous, predatory, and other dietary groups with the two-ratio branch model (model 2) of PAML. Background dN/dS ratios were evaluated with a one-ratio branch model (model 1) of PAML. The distribution density and a box plot of dN/dS values (dN/dS<1) in different dietary groups are displayed. The mean dN/dS value on the branch of the fat-feeding group was significantly higher than that of the background and other groups (***$P < 0.01$). Error bars represent standard deviation (SD). KEGG Kyoto Encyclopedia of Genes and Genomes, PSG positively selected gene, PAML phylogenetic analysis by maximum likelihood.

**Identification of positively selected genes.** A total of 9,306 gene families were identified by OrthoFinder, and 4,546 single-copy orthologous genes were generated to perform positive selection analysis, which annotated at least 60% of the sixteen genomes. The sequences of orthologous genes were aligned by MAFFT software (version 7.455), and a gene tree was generated from the species tree based on the remaining species. The branch-site model in PAML was used to detect PSGs along specific lineages with model A (model = 2 and NSsites = 2). A likelihood ratio test (LRT) was performed to compare model A (sites under positive selection in the foreground; fix_omega = 0) with the null model (sites might evolve neutrally/under purifying selection; fix_omega = 1 and omega = 1) by the codeml program in PAML. The *P* values from the LRTs were evaluated via chi-square statistics, a false discovery rate (FDR) was calculated, and the cut-off was set to 0.1. Sites with a Bayesian empirical Bayes (BEB) posterior probability greater than 95% were selected as positively selected sites in PAML. KEGG and GO enrichment analyses of PSGs were performed by KOBAS 3.0[68] with cut-offs of a *P* value < 0.05 in Fisher's exact tests and an FDR-corrected *P* value < 0.1. *Drosophila melanogaster* genes were selected as the background set. Bubble diagrams of significantly enriched KEGG and GO terms were generated using the R ggplot package.

**Evolutionary rate of fat metabolism.** To investigate the evolutionary rate of fat metabolism among mites with different diets, 122 genes involved in lipid metabolism from the KEGG database (http://www.genome.jp/kegg/catalog/org_list.

html) were included for analysis. The branches were divided into five groups with different dietary types. The dN/dS ratios of selected groups were calculated by the codeml program with the two-ratio branch model (model = 2) in PAML, and the dN/dS values of the background were detected with the one-ratio model (model = 0, NSsites = 0). The density distribution of dN/dS values and a violin plot of dN/dS values for different dietary groups were generated. The dN/dS ratios were compared with that of background data with Student's t test.

**Gene expression analysis.** Sixteen samples from four populations of spider mites were downloaded from the SRA database (BioProject: PRJNA610897)[69], and Trimmomatic (version 0.36) was applied to perform quality control, including cutting adaptor sequences and removing low-quality reads, with the parameters "LEADING:3 TRAILING:3 SLIDINGWINDOW:4:20 MINLEN:45". The qualified reads were mapped to the spider mite reference using HISAT2 (version 2.1.0). Fragments per kilobase of transcript per million mapped reads (FPKM) values were calculated by Cufflinks[70] (version 2.2.1). Then, gene expression in all sixteen samples was sorted according to the median FPKM values. The 200 and 50 most highly expressed genes were selected and analyzed. A heatmap was drawn based on the 50 most highly expressed genes using Heatmapper[71]. KEGG and GO enrichment analyses were performed by KOBAS 3.0 with cut-offs of a *P* value < 0.05 in Fisher's exact tests and an FDR-corrected *P* value < 0.1. Bubble diagrams of significantly enriched KEGG and GO terms were generated using the R ggplot package.

**Serpin family analysis**. The serpin family was manually annotated using methods described in previous studies[72]. The serpin protein sequences of *Ixodes scapularis*, *Metaseiulus occidentalis*, *Dermanyssus gallinae*, *Tropilaelaps mercedesae*, *Varroa destructor*, *Varroa jacobsoni*, *Sarcoptes scabiei*, *Psoroptes ovis*, *Leptotrombidium delicense*, *Dinothrombium tinctorium*, *Tetranychus urticae*, and *Stegodyphus mimosarum* from the NCBI database were used as query sequences. Then, the reference sequences were aligned to the sixteen genomes using BLASTP (version 2.2.29+). Blast hits belonging to the same query protein were combined into one predicted gene from each genome. Then, gene structure was predicted by Gene-Wise, and domains of sequences were estimated based on the Pfam database. Finally, a total of 2,748 positions in the serpin genes of sixteen species were aligned with ClustalW and then applied to construct a maximum likelihood tree in MEGA[73] (version 10.0.1) with 1,000 bootstrap replicates. The resulting tree was visualized in Figtree (version 1.4.3).

**Reporting summary**. Further information on research design is available in the Nature Research Reporting Summary linked to this article.

## Data availability
The final dataset is available at Mendeley Data (https://data.mendeley.com/datasets/xm23f9mkdx/1).

## Code availability
Analysis code is accessible through github (https://github.com/offlinecat/mite_diet).

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

## Acknowledgements
This work was supported by the National Natural Science Foundation of China (31601839). We thank Chunqiao Liu, Yu Liu and Bin Zou for their help during data analysis, and Dr. Shenghan Gao for his advice in the earlier version of the manuscript. We also thank editors and three anonymous reviewers whose feedback helped to improve the manuscript substantially.

## Author contributions
L.W. and D.C. conceived the study; Q.L. supervised the overall data collection and analysis; Y.H.D. and A.S. supervised and performed the comparative genomics pipeline; Y.F.X. provides key discussion; L.Q. and D.C. drafted and revised the manuscript. All authors read and approved the final version of the manuscript.

## Competing interests
The authors declare no competing interests.
