## [Transparent Peer Review File · Communications Biology]

Reviewers' comments:

Reviewer #1 (Remarks to the Author):

This manuscript studied the natural selection patterns of mites with different feeding habits through comparative genomics method. They identified the positively selected genes and gene family expansion related with herbivorous, blood-feeding, and fat-feeding habits. These potential candidate genes provide insights into the dietary evolution of mites under different selection pressures. I have some comments for improving the manuscript.

1. Line 98: how many genes were used for phylogenetic tree construction?
2. Line 99: should be "relaxed".
3. Line 114: "detected to be common in the groups of ...".
4. Line 136: should be "correlated with".
5. Line 138: "the number of PSGs (more than 20) was much higher....".
6. Lines 140-141: I doubt whether there were more than 200 sites under positive selection pressure for one gene. So, please check the DNA sequence alignments of this gene among these species, to avoid the sequence misalignment.
7. Lines 163 and 189: "owing to" should be "belonging to"?
8. Lines 167 and 209: please change as "manual filtration".
9. Line 193: please delete "cleverly".
10. Lines 205 and 252: When using the PZ and AA firstly, please give its full name.
11. Lines 249-251: please give related reference to support the absence of FAAH2 in rats and mice.
12. Line 262: "any expansion gene families"??
13. Line 263: What do you mean about "genetic expansion"?
14. Lines 274-275: What do your mean about "Different levels of genetic convergence under the same diet ¹_{SEP} was identified"?
15. Figure 1: the species Latin names should be italic.
16. Figure 3A and 3B. It seemed that corrected P-values were not insignificant. How do you think about the results?
17. Figure 3D: the figure is unclear.

Reviewer #2 (Remarks to the Author):

This study provides the genetic evidence of mites and ticks adaptation to different dietary habits based on the comparative analysis of genomes of 16 arachnids. Authors analyzed mites and ticks of adaptations to Herbivory, blood-sucking and Fat-feeding by using the methods such as gene family expansion and gene selection pressure.

The finding of this working is interesting, but I got some comments as below:

Line 89. The genome assemblies in Table 1 used in this analysis should be cited clearly and separately, so that the other researcher can find correct paper resources to repeat this work.

Line 90. Authors should submit the gene annotations for the 5 reannotated genomes to public database.

Line 93. In this manuscript, the authors expected to use the annotated gene sets, especially the protein-coding gene sets, to understand the mite evolution. However, only the genome assembly qualities of the selected mite and tick genomes were tested with BUSCO. To exam the qualities of annotated gene sets in this mite and tick genomes apparently more important. Authors should also use the BUSCO to evaluate the qualities of annotated gene sets.

Line 100. There was a long-time debate for the Acari is diphyletic or monophyletic. It would be good to discuss this based on figure 1.

Line 166 and 367. Authors should provide the source of RNAseq data for the four spider mite populations.

Line 167,168 and 172. How were the top 200 genes found based on Sixteen samples from four population of spider mite? Please the authors provide the method and the gene list. Which method did authors used for GO and KEGG enrichment for the expanded genes? Authors should clarify in the method part.

Line 174. How were the top 50 genes found based on Sixteen samples from four population of spider mite? Please the authors provide the method and the gene list.

Line 211. Please provide high resolution Supplementary Fig. 4. It is not readable. Authors should label the bootstrap numbers on the tree, and also provide gene alignment file and the tree construction method in the method part.

Line 332. For the KEGG and GO enrichment analysis, please authors explain why they selected the far related *Drosophila melanogaster* as background set. The Kobas used in this enrichment analysis supports a wide range of species, including mite and tick models species, such as *Ixodes scapularis* and *Tetranychus urticae*. The insects are innately very different from the mites and ticks in some pathways. Please authors explain how the gene enriched with the *D. melanogaster* gene sets could reflect the mite and tick biology? What is the p-value or q-value cutoff for Kobas result?

Reviewer #3 (Remarks to the Author):

In this study, the authors compared genomes of 16 Arachnida species, and showed different patterns of diet adaptations. I have a few concerns:

1. The manuscript was mainly based on published genome data. Expression data of spider mites were also used. It is not clear why only expression data of herbivory species were used. It seems the authors are choosing data arbitrary. I think they should provide more detailed and scientific explanations about data selection.

2. The authors showed that Arachnida species with different diet showed different evolutionary patterns based on some regular analyses. These results are expectable. It is not clear what is the key scientific problem the authors want to solve based on their analyses. What is the purpose of doing these analyses as well as publishing a paper, and how these results will serve the purpose? In the abstract and the conclusions, the authors mentioned twice: "These different genetic bases provide a new perspective for the study of the evolution and diversification of this group, and offers potential drug targets for pest control." These statements are repetitive, and are very vague. What is the "new perspective" and how to offer potential drug targets? Details should be discussed at least. In addition, not all mite species analyzed in this study are pests. For example, *Metaseiulus occidentalis* is a predatory mite natural enemy.

3. There are many typos and formatting errors in manuscript (including the references). I think the authors should pay more attentions to fix these problems.

Response letter COMMSBIO-20-3585

Genomic implications in diet evolution: Comparative analysis of mite genomes reveals positive selection for diet adaptation

Dear Reviewers,

Many thanks for your important and helpful suggestions for our manuscript entitled “Genomic implications in diet evolution: Comparative analysis of mite genomes reveals positive selection for diet adaptation” (COMMSBIO-20-3585). In the following responses, we have carefully addressed all the issues, and we have revised our manuscript accordingly. Our references to line numbers refer to the no markup view that we have uploaded as a ‘Acari_Diet_maintext_changes_tracked.docx’ file. All changes have been accepted in the clean revised manuscript uploaded as a ‘Acari_Diet_supplementary_maintext_clean.docx’ file. We hope you find that we have adequately addressed all of the suggestions and that our manuscript is now suitable for publication. Please let us know if you have any further questions or suggestions.

Our point-by-point responses are as follows:

Reviewer #1 (Remarks to the Author):

This manuscript studied the natural selection patterns of mites with different feeding habits through comparative genomics method. They identified the positively selected genes and gene family expansion related with herbivorous, blood-feeding, and fat-feeding habits. These potential candidate genes provide insights into the dietary evolution of mites under different selection pressures. I have some comments for improving the manuscript.

Thank you for your recognition! We appreciate your positive comments and we made corresponding revisions to your suggestions.

Comment (1) Line 98: how many genes were used for phylogenetic tree construction

A total of 17,910 homologous genes were identified. Due to the different techniques of sequencing and assembly of the 16 genomes, critical filtration has been carried out in the species selection (lines

285-291) and reannotation was conducted for 6 genomes. However, the discrepancy of the amount of gene information cannot be avoided and the number of single-copy homologous genes annotated is limited. Finally, 65 single-copy homologous genes with 48,831 nucleotides were annotated by all 16 species and applied to construct the species tree.

Comment (2) Line 99: should be “relaxed”.

Thank you for your suggestion. We have corrected it in line 101.

Comment (3) Line 114: “detected to be common in the groups of ...”.

Thank you for your suggestion. We have corrected it in line 115.

Comment (4) Line 136: should be “correlated with”.

Thank you for your suggestion. We have corrected it in line 130.

Comment (5) Line 138: “the number of PSGs (more than 20) was much higher....”.

Thank you for your suggestion. We have corrected it in line 140.

Comment (6) Lines 140-141: I doubt whether there were more than 200 sites under positive selection pressure for one gene. So, please check the DNA sequence alignments of this gene among these species, to avoid the sequence misalignment.

Gene *CES2* (ID: OG0000064) is an outlier in blood-feeding group shown in Fig1D with a total of 1,128 amino acid sites after deleting aligned gaps. 285 positively selected sites were found, among which 197 sites with a posterior probability of over 95% were detected. We haven't found any errors in its result of sequence alignment (See following figure 1).

Psoroptes_ovis hIfGtdhnffngpiEiyksIrYAnvtedYDAidyrE1CmQhpfkfmSEnCLYLNiWfPfr
 Metaseiulus_occidentalis qmRGniKTsaGvrvnSFLGIPYAqPVGvFEAKK1PKSCpQpnvsDtSEDCLYLNiWfPfc
 Tropilaelaps_mercedesae qmKGifKTssGirVnSFLGIPYAqPtnvFEAKK1PKSCpQpnisDtSEDCLYLNiWfPfc
 Varroa_destructor qmKGifKTssGirVnSFLGIPYAqPInvFEAKK1PKSCpQpnmsEtSEDCLYLNiWfPfc
 Varroa_jacobsoni qmKGifKTssGirVnSFLGIPYAqPInvFEAKK1PKSCpQpnmsEtSEDCLYLNiWfPfc
 Dermanyssus_gallinae vVRGvvsSrmGrtVEAFLaTPfaPIGqFDATRakpgCmQlnndDttEDCLhLNvfvPrt
 Ixodes_scapularis lVaGtrievgdntVDAFLGIPYAePVGtYnATskPKACwQlsysNaSEDCLYLNvWkPvs
 Brevipalpus_yothersi vlRGiiQTptGkyVDAFLGIPYAaPtGeYDATKfsgpCyQlpnvpvdEDCLtLNiWvPkk
 Tetranychus_urticae yVRGrvvSptGkpVDAFLGIRYAKPtGifnATsfsgACyQvnpvpldEDCLsvNiWvPrp
 Dermatophagoides_farinae mVRGqvRSlnGrlVDEFLGIPfAKPIGifnATQkPKSCyQvdqtNldEDCLtLNiWvPyp
 Hypochthonius_rufulus lVRGvLRaatGkpVdVfyGIPYAKPIGifnAshrPnSCfQiantplSEDCLkiNvWvPhp
 Stegodyphus_mimosarum pVRGvmlSghGkeVdVFLGIPYAKPVGvlnATEaPnSCvQiansplSEDCLtvsVvPkp
 Steganacarus_magnus qfnGfniiigskqVdiFLGIPYAKPIGpiDATRwPnpCiqrhntNfSEDCLYLNiWtPnd
 Dinothrombium_tinctorium tTeGkvEyrvGkiatFLGvPYAKaIGiYrATvekpAClQwhavNdSEDCLYLNiYaPie
 Leptotrombidium_deliense lTeGfkKnlfgnyVntFLGIPYAqPIntlqAmKlknClQylthemSEDCLYLNvWtksv
 Sarcoptes_scabiei rLRGfvQnvtGveVqtFLSIPYAqPIGtrDATRiPppCfQplneNitEDCLYLNvSPln

 Psoroptes_ovis eielKPVIVyIHGGGFqangINrlpsDCSgyvdKGDVVFVSIqYrvGaLGLFYtGdrraP
 Metaseiulus_occidentalis dgRsrPVIVyIHGGGFihGgtNwpIFDGAeLAAkaetVVVSI1NYRLGaLGLFYvpSnper
 Tropilaelaps_mercedesae dgRsrPVVvYIHGGGFihGgINwpIFDGAeLAGKGDVVVtVNYRLGaLGLFYipSnper
 Varroa_destructor dgRsrPVVvYIHGGGFvYgVNwpIFDGAeLAGKGDIVVtVNYRLGaLGLFYipSnper
 Varroa_jacobsoni dgRsrPVVvYIHGGGFvYgVNwpIFDGAeLAGKGDIVVtVNYRLGaLGLFYipSnper
 Dermanyssus_gallinae adRpKPVfVlylGGaFlwGenqLaYDGVefAAQaDaifVaVNYrvnmfG1LnsnSsDiP
 Ixodes_scapularis scKksPVIifIHGGaFqwGdsalFLYDpanfvAlsDVVfvtfNYRLslLGLFslsTaEmP
 Brevipalpus_yothersi rrsngacLlWlYGGGFstGttsLdVYNCSiLAGeEsIiViSVNYRLasLGFiyIIGredaP
 Tetranychus_urticae rpKnsaaVLWlYGGGFsGSSsLdfYDGSvLAGeEsIifVSINyrvasLGFiffdTsDaP
 Dermatophagoides_farinae rpKnsaaVLWlYGGGFysGSSsLdLYDgklfAtEenIifVSINyRLnLGLFYfEntDiv
 Hypochthonius_rufulus rpKnsaaVLWlYGGGFysGSSsLdLYDskilAtEenIiVSmNYrvanLGLFFeNaEaP
 Stegodyphus_mimosarum rpesaaVLWVfyGGGFysGttdLdVYDpkiLcSEeEiIVSINyrvasLGLFYfdrpDvP
 Steganacarus_magnus sallKPVlVWlIHGGglivGSSnMiItsGctLAAKGDVVVVSINyRLdyfGFfYsGtdDaP
 Dinothrombium_tinctorium trRpKtVMWlIHGGaFlrGfINfhYDGAmlsAfGevVVVfvVsYRLGimGFmsaGekqlP
 Leptotrombidium_deliense nkammPiMiWlHGGGFteGSVNIIdeniGevLAlYGDVVVtfNYrvGaLGLFdlGlaEsP
 Sarcoptes_scabiei grKntntVMiWlIHGGlFtiGSgtIdeYDGrmlAAfGnViVVSIqYRLGvfgLdldTeriP

 Psoroptes_ovis GNqLALKWiekNIKSFGGDPtsvTLmGESAGswaIsalILSPmaKGLFhRAILhSGSiiE
 Metaseiulus_occidentalis aNmLALRWvksNIKAFGGDhNKITimGQdAGsaSVGyHILtPrStGLFKRAvmQSGSPFS
 Tropilaelaps_mercedesae aNmLALRWvksNIKAFGGDhNKITvmGQdAGAaSVGyHILtPrStGLFKRAvmQSGSPFS
 Varroa_destructor aNmLALRWvksNIKAFGGDhSKITvmGQdAGAaSVGyHILtPrStGLFKRAvmQSGSPFS
 Varroa_jacobsoni aNmLALRWvksNIKAFGGDhSKITvmGQdAGAaSVGyHILtPrStGLFKRAvmQSGSPFS
 Dermanyssus_gallinae GNmMALlVWRDNIaAFGGnPDvTLgGQSAGafSaaIHsySPvSKGLFKRAvLmSGts1S
 Ixodes_scapularis GNvLvLKWVRkNIESFGGDPkevTLsGQSAGiSaGLHaiSphsQLGFKRvvmQSGtPFS
 Brevipalpus_yothersi GNAMaIeVWRDNIaAFGGnPanITLFCESAGaSaIafHLLSP1SRnLFsqgILQSGSstS
 Tetranychus_urticae GNAMameWiRENIaAFGGnPanITiFCESAGaSaLHLLSP1SRnvFsqAILQSGSatic
 Dermatophagoides_farinae GNAMALqWihTNIeKFGGnrHnITLFCESAGaSaIafHLLSP1SghLFnqAILQSGaPtv
 Hypochthonius_rufulus GNAMALqWVkdNIafFGGnPNnvTLFCESAGaSaVGLLSP1SRhLFsqAILeSGGpTv
 Stegodyphus_mimosarum GNAMgLeWirdNIayFGGnPHnvTLFCESAGaSaVGLHLLSP1SRnLFsqAILmQSGSatS
 Steganacarus_magnus GNALALKWvndNvEYFGGDPNRITiFGaSAGgwSVsLlLvSPitRhLyKnAvvmSGSahv
 Dinothrombium_tinctorium GNlFALKWIRaNIYAFGGDPNKIvLFGQgsGAvSIGyHLiSP1SghLFqRAImQSGnPlS
 Leptotrombidium_deliense GNqtAfRvWkDnahSFGGDPNsITLFGvSsGAiaIGLHMvSPeSKhLFnRAILQSGSPv1
 Sarcoptes_scabiei GNmtAikWVRqNIRfFGGDPksITLFGtSAGsiSIGfHMfSP1tKhLfkRAILQSGSPmi

Psoroptes_ovis ykyspDyAlektikIglennCIDClgKvspykLaSkliKllegkFtgLIyLPgQinpvDL
 Metaseiulus_occidentalis fvntkDQgetLfrsLAsytdClqCmKKQpfenTAAasdKfgansFfPIMGiPKRFsnvDL
 Tropilaelaps_mercedesae fvntkDQgetLfrsLAgytdCvtCmKR1PfesIISaseKfgansFfPILGiPKRFsnvDL
 Varroa_destructor fvntkDQgetLfrsLAsytdClqCmKK1PfesIITaseKfgansFfPILGiPKRFsnvDL
 Varroa_jacobsoni fvntkDQgetLfrsLAsytdClqCmKK1PfesIITaseKfgansFfPILGiPKRFsnvDL
 Dermanyssus_gallinae lkfeqtpAedLlaaIAleLdCakCitakPltharl1EvKrmwfnFapHaGLPiKVhveDI
 Ixodes_scapularis ligmahKgakfinaAgtLGCLvCLRKldaksMfki1qsavqqfFapVIGfPKm1RfkDI
 Brevipalpus_yothersi pwidrtEAFrrsltLAKnIGCThCLQeadpiqLLTnEppvdfafVpIVdldKnFKqTnL
 Tetranychus_urticae pwsdrkKAYqrs1aLAqaVGCIECmQsiPaseLVAqEttvefaFiPIVdldKnFKkTnL
 Dermatophagoides_farinae pwwdkKihqrglmLAesVGCIDCLRqtdpalLMqnEstdvfaFtPILdltKRFKkTnL
 Hypochthonius_rufulus swgdhkEmt1rglrLAeaVGCLECLRKadpvdLVTnEsgnvefpFiPVVdldKnFKkTnL
 Stegodyphus_mimosarum pwwdrkEnmrrgllLAeaLkCvDCL1RkdpfeMVSNEwgnvefpFapVVDldKnFKkTnL
 Steganacarus_magnus mykdaklAlnsytevAkkLsCfECLRNvaaqdfMSdKfdkslteirvlyGLPgdcRnanv
 Dinothrombium_tinctarium ptlgsgsApyklekvAkrsCCIDCLRqvsansLl1fQeElidtvFyPIEdlnqKFSqqEv
 Leptotrombidium_deliense vnsyysnSanVaqvAevLqCLkCLRkskatqfLeaqrQissftFtPspyLPKvvsdaqv
 Sarcoptes_scabiei lkdalERgekLaekfAslIGCVDCidRtPikllyeaQnElnvpvFmPtIriinlFKlqtK

Psoroptes_ovis ivGhtysEygFyLaitnFl1llkhhkDrwqrifKhYtrfeipsvyhAkIfkEYaktNrftd
 Metaseiulus_occidentalis LLSItKsEGaFFLqHflspFtnvnDNpgEfyLRvF1SallggkptAsLne1TqntePetq
 Tropilaelaps_mercedesae LvsTaKsEGaFFLqHflspFtnvaDNpgEfyLRvF1SallggkptAsLneFTkntePrte
 Varroa_destructor LvsTaK1EGaFFLqHflspFtniaDkpeEfyLRvF1SallggkptAsLinqYTqntePktD
 Varroa_jacobsoni LvsTaK1EGaFFLqHflspFtniaDkpeEfyLRvF1SallggkptAsLinqYTqntePktD
 Dermanyssus_gallinae mLGmsQrEGdFlfhafrewipgladNspEttLRlaikmlfgvhrdtaklYkaylNadDr
 Ixodes_scapularis iLGTNRDECT1Fvdn1rsqiptlgtlatd1avtvv1Apmfdisisqsrri1vayfggdDt
 Brevipalpus_yothersi LvGSNhDEGTyVlvVhePk1fnltESraEgyvKklfAdnpwiiqEAiMqeYTPw1NPdne
 Tetranychus_urticae LtGSNRDEGTyVFLVhsPhifnlsESrsEslIRIiyphlsplaqEAvIqeYThwiNPdDe
 Dermatophagoides_farinae LLGnNKEGTyFiiYklnldifkfeEtradksvRlltpnmhpigqsAiIyeYTdw1NPdDk
 Hypochthonius_rufulus LLGSntEGCTFFiiYhltelfkkvEtredekavKdl1npyvgkvg1DAvIfeYTdw1NadDk
 Stegodyphus_mimosarum mtGSNsEGTYFivYltdYfknqENrqEhavKelnpyvgelaqEAiIfqYTdweNPYDk
 Steganacarus_magnus LLGTtdDEGsvFLhdnityalln1pkalvEkifslFgSlanahenahkLyisqspeNsyDl
 Dinothrombium_tinctarium LMGrNsNEGsFyvwlalFflyldsktieEdeLKEFgprkarqyyEh1f1psilskfkpni
 Leptotrombidium_deliense LMG1tKDEas1FLhFhsEveftnkstleEemfvdvkrnhvhtqah1Lat1llkessnEd
 Sarcoptes_scabiei kkGfNqNEGalmLhlsyPqFylrdkkledkkMiemgSglsemaastian1FikgkktDsr

Psoroptes_ovis	vRRiLiEflsDYrnlCPTliEgqEYsNtlkrVYafRFDHySpfcEWMkscHfidlpymMG
Metaseiulus_occidentalis	f1RnISaVIGDYpylCaItEFgtEYANlrHNVYhmqyDHRpwhPtWfpstHnddImFWLG
Tropilaelaps_mercedesae	f1RnISaVIGDYpFvCaItEFgtEYANlrHNVYhmqFDHRpwhPtWfsatHfdDvmFWLG
Varroa_destructor	f1RnISaVIGDYpFvCaItEFgtEYANlrHNVYhmqFDHRpwhPtWfsatHfdDvmFWLG
Varroa_jacobsoni	f1RnISaVIGDYpFvCaItEFgtEYANlrHNVYhmqFDHRpwhPtWfsatHfdDvmFWLG
Dermanyssus_gallinae	isRigSdIVGDtvFdCPvqfFAehaASqGvrVrhVvyaHsptnlnlgeptHsdDlpFmLG
Ixodes_scapularis	vevifSKIIGDaaFnCPTklFAadiaASqG:NtYrYlFtHRpsWPKWLGvaHtDEIvFthG
Brevipalpus_yothersi	nRdkIdKIVGDYhFaCPgiEmAhrYafyGNNVwnYyFtHRSsWPsWMGViHaDEImFllG
Tetranychus_urticae	nReatKfVGDYhFTCPvnEmsyrYAlYGNdVwtYhFtHRSsWPsWMGViHgeEIkFvLG
Dermatophagoides_farinae	nRdaIdKIVGDYhFvChvnrKFAdrYASvGNeVYmYyFtHRSsWPKWGMtiHgDEIpFifG
Hypochothonius_rufulus	nRdaIdKIVGDYhFTChvnrElAyrYsaaGNdVYmYyFtHRSsWPKWGMmHaDEInFvfG
Stegodyphus_mimosarum	nRdaIdKIVGDYqFTCSvnrEiAlrYAetGNdVYmYyFtHRSsWPKWGMmHaDEInFifG
Steganacarus_magnus	iRRsIgtatIGDYmlCPTlhlAkalfgnatVYqYlFsgaknansWhGsdHysdVvyWfG
Dinotrombium_tinctorium	aRKeLeLlIqkvfFTCPemKFidaFtNaGkrVYyYelaHeteWPKsMGskHfDEIqyifG
Leptotrombidium_deliense	yvamLhtalsDlaFnCPsiiFAeElAkanktVYfyvYhKsafadWMGVpHlslmpyvfg
Sarcoptes_scabiei	waQkVSdVfsDvmFvCPTyQFidKlqSfhNqiYlYlFgqRakWgdWmdVtHqDEmNFvMG

Figure 1. Protein sequence alignments of carboxylesterase (After deleting the gaps)

Comment (7) Lines 163 and 189: “owing to” should be “belonging to”?

Thank you for your suggestion. We have revised it to “Cyp18a1, a cytochrome P450 enzyme” in line 165.

Comment (8) Lines 167 and 209: please change as “manual filtration”.

Thank you for your suggestion. We have corrected it in line 168 and line 212.

Comment (9) Line 193: please delete “cleverly”.

Thank you for your suggestion. We have corrected it.

Comment (10) Lines 205 and 252: When using the PZ and AA firstly, please give its full name.

Thank you for your suggestion. *Arachidonate acid* (AA) was first mentioned in line 248. *Protein Z* (PZ) was supplemented in the revised manuscript, see line 209.

Comment (11) Lines 249-251: please give related reference to support the absence of FAAH2 in rats and mice.

Thank you for your suggestion! We have added this reference no.62 in the revised manuscript.

Comment (12) Line 262: “any expansion gene families”??

We are sorry for the confusing expression. We have revised it to “no common expanded gene family was found in this group” (see line 151).

Comment (13) Line 263: What do you mean about “genetic expansion”?

We are sorry for the confusing expression. We have revised it to “expanded gene families” in line 150.

Comment (14) Lines 274-275: What do your mean about “Different levels of genetic convergence under the same diet ¹¹¹_{SEP} was identified”?

We are sorry for the confusing expression. We have deleted the sentence.

Comment (15) Figure 1: the species Latin names should be italic.

Thank you for your suggestion! We have revised it in Figure 1.

Comment (16) Figure 3A and 3B. It seemed that corrected P-values were not insignificant. How do you think about the results?

Thanks for raising this issue! To find the significant pathways and gene ontology, we set the cutoff of P value < 0.05 in Fisher's exact test and FDR-corrected P < 0.1 according to the thresholds recommended in the previous studies (Liu, Yao-Zhong et al., 2017; Hulsege, I et al., 2017). The KEGG pathway and Gene ontology related to detoxification function mentioned in the lines 165-181 (the following table 1) were considered statistically significant according to the thresholds.

Term	Database	P-Value	Corrected P-Value
Response to stimulus	Gene Ontology	2.84E-05	0.0031299
Response to external stimulus	Gene Ontology	3.93E-05	0.0037041
ABC transporters	KEGG PATHWAY	0.0054869	0.0910888

1. Liu, Yao-Zhong, Lei Zhang, Astrid M. Roy-Engel, Shigeki Saito, Joseph A. Lasky, Guangdi Wang, and He Wang. "Carcinogenic effects of oil dispersants: A KEGG pathway-based RNA-seq study of

human airway epithelial cells." *Gene* 602 (2017): 16-23.

2. Hulsege, I., A. Kommadath, and M.A. Smits, Globaltest and GOEAST: two different approaches for Gene Ontology analysis. *BMC Proc*, 2009. 3 Suppl 4: p. S10.

Comment (17) Figure 3D: the figure is unclear.

Sorry for the unclear version. We have split Figure 3 into new Figure 3 and 4 to show the information of figures better.

Reviewer #2 (Remarks to the Author):

This study provides the genetic evidence of mites and ticks adaptation to different dietary habits based on the comparative analysis of genomes of 16 arachnids. Authors analyzed mites and ticks of adaptations to Herbivory, blood-sucking and Fat-feeding by using the methods such as gene family expansion and gene selection pressure.

We appreciate your positive comments and we have made corresponding revisions to your suggestions.

The finding of this working is interesting, but I got some comments as below:

Comment (1) Line 89. The genome assemblies in Table 1 used in this analysis should be cited clearly and separately, so that the other researcher can find correct paper resources to repeat this work.

Thank you for your suggestion! We have supplemented separate citations for each genome in the revised manuscript (Table 1). Researchers can also obtain the genome data through the GenBank Assembly Accession ID in Table 1.

Comment (2) Line 90. Authors should submit the gene annotations for the 5 reannotated genomes to public database.

Thank you for your suggestion! We have uploaded the annotation files of the reannotated genomes to Mendeley Data (<https://data.mendeley.com/datasets/xm23f9mkdx/1>). This information is noted in the line 404-406 in the revised manuscript.

Comment (3) Line 93. In this manuscript, the authors expected to use the annotated gene sets, especially the protein-coding gene sets, to understand the mite evolution. However, only the genome assembly qualities of the selected mite and tick genomes were tested with BUSCO. To exam the qualities of annotated gene sets in this mite and tick genomes apparently more important. Authors should also use the BUSCO to evaluate the qualities of annotated gene sets.

Thank you for your suggestion! We have finished the analysis and provided the BUSCO results in Supplementary Table 1. We observed high BUSCO scores of genome completeness (average 90.9%) and gene completeness (average 79.9%). The proportion of complete gene is not very high in species *Steganacarus magnus* and *Hypochothonius rufulus*. However, the two species were set as background species when conducted the positive selection analysis and would not affect the identification of positive selection signals.

Comment (4) Line 100. There was a long-time debate for the Acari is diphyletic or monophyletic. It would be good to discuss this based on figure 1.

Thank you for your suggestion! The debate about the monophyly of Acari has always been a hot topic. A recent study indicated that Acari constitutes a monophyletic group (Shown in the figure below) nested within a monophyletic Arachnida based on transcriptomic data from 95 species. (Jesus Lozano-Fernandez, *et al.*, 2019). However, the study published at the same year recovered the major mite lineages by using ultraconserved genomic elements (UCEs) and found mites to be non-monophyletic (Van Dam, 2019). Our data was insufficient to address this problem, unless we add the groups that can represent all the spiders of Arachnida, such as whip spiders, llamshade spiders, mygalomorph spiders, hooded tickspiders, Sun Spiders, and so on. However, it is not the main focus of our study. It is a good idea for our follow-up research.

1. Lozano-Fernandez, Jesus, et al. "Increasing species sampling in chelicerate genomic-scale datasets provides support for monophyly of Acari and Arachnida." *Nature communications* 10.1 (2019): 1-8.

2. Van Dam, Matthew H., et al. "Advancing mite phylogenomics: designing ultraconserved elements for Acari phylogeny." *Molecular ecology resources* 19.2 (2019): 465-475.

(Jesus Lozano-Fernandez, et al., 2019)

(Van Dam, 2019)

Comment (5) Line 166 and 367. Authors should provide the source of RNAseq data for the four spider mite populations.

Thank you for your suggestion! RNAseq data was from the SRA database (Bioproject: PRJNA610897; Xue, Wenxin, *et al.*, 2020). We have supplemented this information in the Method in the revised manuscript (see line 375).

1. Xue W *et al.*, "Geographical distribution and molecular insights into abamectin and milbemectin cross-resistance in European field populations of *Tetranychus urticae*.", *Pest Manag Sci*, 2020 Aug;76(8):2569-2581

Comment (6) Line 167,168 and 172. How were the top 200 genes found based on Sixteen samples from four population of spider mite? Please the authors provide the method and the gene list. Which method did authors used for GO and KEGG enrichment for the expanded genes? Authors should

clarify in the method part.

Thanks for raising this issue! First, we conducted short sequence mapping using HISAT2 and calculated Fragments per kilobase per million mapped reads (FPKM) values for genes of each sample by Cufflinks (Version 2.2.1). Then, genes present in all sixteen samples were sorted by the median FPKM value. Then the 200 most highly expressed genes were selected and analyzed. KEGG and Gene ontology enrichment analysis were performed by KOBAS 3.0 with the cutoff of P value < 0.05 in Fisher's exact test and FDR-corrected $P < 0.1$ according to the thresholds recommended in the previous studies (Liu, Yao-Zhong et al., 2017; Hulsegge, I et al., 2017). The bubble diagrams of significantly enriched KEGG and Gene ontology were generated using R ggplot package. We have supplemented details in the Methods section (lines 374-387) and provided the gene list in the supplementary Table 8 in the revised manuscript.

1. Liu, Yao-Zhong, Lei Zhang, Astrid M. Roy-Engel, Shigeki Saito, Joseph A. Lasky, Guangdi Wang, and He Wang. "Carcinogenic effects of oil dispersants: A KEGG pathway-based RNA-seq study of human airway epithelial cells." *Gene* 602 (2017): 16-23.

2. Hulsegge, I., A. Kommadath, and M.A. Smits, Globaltest and GOEAST: two different approaches for Gene Ontology analysis. *BMC Proc*, 2009. 3 Suppl 4: p. S10.

Comment (7) Line 174. How were the top 50 genes found based on Sixteen samples from four populations of spider mite? Please the authors provide the method and the gene list.

The methods of selecting the 50 most highly expressed genes were consistent with those of selecting the 200 most highly expressed genes. We have supplemented details in the Methods section (lines 374-387) and provided the gene list in the supplementary Table 8 in the revised manuscript.

Comment (8) Line 211. Please provide high resolution Supplementary Fig. 4. It is not readable. Authors should label the bootstrap numbers on the tree, and also provide gene alignment file and the tree construction method in the method part.

Sorry for the unclear image and statement. We have replaced Supplementary Fig. 4 with a version of higher resolution and provided the detailed information of tree construction in the Method section

in the revised manuscript. Supplementary Fig. 4 has been changed to Supplementary Fig. 3 in the revised manuscript. Since the tree figure focuses on displaying the expanded gene number and clades, we have provided the bootstrap value in a separate tree file instead of the tree figure to avoid shading the 212 branches. We have submitted this tree file with bootstrap value and gene alignment file to the Mendeley Data (<https://data.mendeley.com/datasets/xm23f9mkdx/1>) due to the big size of the files. We have provided a “Data availability” section in the revised manuscript (lines 404-406).

Comment (9) Line 332. For the KEGG and GO enrichment analysis, please authors explain why they selected the far related *Drosophila melanogaster* as background set. The Kobas used in this enrichment analysis supports a wide range of species, including mite and tick models species, such as *Ixodes scapularis* and *Tetranychus urticae*. The insects are innately very different from the mites and ticks in some pathways. Please authors explain how the gene enriched with the *D. melanogaster* gene sets could reflect the mite and tick biology? What is the p-value or q-value cutoff for Kobas result?

Thanks for raising this issue! There are several considerations: (1) *D. melanogaster* is a well-studied specie in KEGG and Gene ontology (GO) in Arthropod, sharing a common ancestor with ticks and mites. (2) KEGG database is available for all species while Gene ontology database is only available for *I. scapularis*, and *D. melanogaster*, not for *T. urticae*. (3) When *D. melanogaster* was the background specie, more gene data of the 200 most highly expressed genes could be obtained from KEGG and GO annotations to retain more information for enrichment analysis. When *D. melanogaster* was set as background set, 60 genes could be obtained from KEGG pathway while 147 genes could be obtained from GO annotation. When *I. scapularis* was set as background set, 53 genes could be obtained from KEGG pathway while 110 genes could be obtained from GO annotation. When *T. urticae* was set as background set, 57 genes could be obtained from KEGG pathway while no information was obtained from GO annotation. Among these genes obtained from KEGG annotation, more than 90 percent genes of *I. scapularis* and of *T. urticae* were annotated in *D. melanogaster*. Among the genes obtained from GO annotation, 88 percent genes of *I. scapularis* were annotated in *D. melanogaster*. Hence, we chose *D. melanogaster* as the background set to generate more information. We set a threshold of P value < 0.05 in Fisher's exact test and FDR-

corrected $P < 0.1$ for KOBAS result.

Reviewer #3 (Remarks to the Author):

In this study, the authors compared genomes of 16 Arachnida species, and showed different patterns of diet adaptations. I have a few concerns:

Comment (1) The manuscript was mainly based on published genome data. Expression data of spider mites were also used. It is not clear why only expression data of herbivory species were used. It seems the authors are choosing data arbitrary. I think they should provide more detailed and scientific explanations about data selection.

Thanks for raising this issue! Our study aimed to explore the dietary adaptations of mites based on released genomes. We have supplemented the details of data selection in the revised manuscript (lines 285-291). “(1) All Arachnida species in the database (before 2020.06) were searched as candidates; (2) Genome with the best completeness score was selected as the representative if there were two or more genomes for one species; (3) Species with genome completeness less than 80% were eliminated; (4) Gene prediction was conducted for the genomes lack of gene annotation information; (5) Contigs less than 1kb were excluded from the whole analysis”. Finally, we chose one tick and fourteen mites, and the velvet spider as the background from 26 candidate species (see the following Table 1).

To see if the results of positive selection and gene expansion have been reflected in gene expression, we supplied transcriptome analysis for the different dietary groups. Sampling of mites is a big challenge, so we conducted database search for mite transcriptome of lipid-feeding, blood-feeding and plant-feeding species. The samples taken from whole blood or whole tissue transcriptome from natural individuals were included. After the database search, two samples of blood-feeding were insufficient to do biological repetition and excluded; four samples of lipid-feeding were included. Only “Ribosome” was statistically significant with correction Fisher test $P < 0.05$ and FDR corrected $P < 0.1$ (see the following Table 2). Finally, we have shown details for gene expression of

the herbivorous group in our manuscript (see line 170-185).

Table 1. Genomes feature of candidates

Species	GenBank Assembly	Number	Total size of	Number	Number	Number	N50 contig	BUSCO completeness
	Accession	of	contigs	of	of	of	length	
		contigs		contigs >	contigs >	contigs >		
				500 nt	1K nt	10K nt		
Achipteria coleoprata	GCA_000988765.1	70,955	87,501,022	41.20%	26.60%	1.50%	3,583	C:82.5%[S:80.5%,D:2.0%],F:9.2%,M:8.3%,n:303
Brevipalpus yothersi	GCA_003956705.1	2,451	70,567,388	99.90%	96.90%	62.20%	56,520	C:83.9%[S:82.2%,D:1.7%],F:3.0%,M:13.1%,n:303
Dermatophagoides farinae	GCA_000767015.1	7,089	51,638,314	91.30%	83.70%	22.50%	14,557	C:90.4%[S:90.1%,D:0.3%],F:3.3%,M:6.3%,n:303
Dermatophagoides farinae	GCA_002085665.1	1,716	91,934,661	100.00%	99.90%	92.00%	188,869	C:95.4%[S:77.6%,D:17.8%],F:0.7%,M:3.9%,n:303
Dermatophagoides pteronyssinus	GCA_003076615.1	1,390	66,623,663	100.00%	100.00%	90.20%	80,070	C:64.7%[S:54.1%,D:10.6%],F:2.6%,M:32.7%,n:303
Dermatophagoides pteronyssinus	GCF_001901225.2	4,324	68,557,481	94.20%	77.30%	25.60%	74,612	C:70.3%[S:68.6%,D:1.7%],F:3.3%,M:26.4%,n:303
Dinothrombium tinctorium	GCA_003675995.1	25,507	180,156,552	92.80%	88.30%	21.50%	16,116	C:80.5%[S:41.9%,D:38.6%],F:4.3%,M:15.2%,n:303
Euroglyphus maynei	GCA_002135145.1	72,786	43,436,854	34.90%	13.10%	0.00%	787	C:66.0%[S:65.3%,D:0.7%],F:20.5%,M:13.5%,n:303
Hypochothinius rufulus	GCA_000988845.1	151,357	171,814,378	39.00%	25.10%	1.30%	3,254	C:89.1%[S:87.1%,D:2.0%],F:5.9%,M:5.0%,n:303
Leptotrombidium deliense	GCA_003675905.2	17,210	15,149,695	59.20%	31.50%	0.00%	1,422	C:84.5%[S:83.5%,D:1.0%],F:6.3%,M:9.2%,n:303
Platynothrus peltifer	GCA_000988905.1	118,520	100,021,536	48.80%	23.10%	0.10%	1,326	C:75.2%[S:74.9%,D:0.3%],F:11.9%,M:12.9%,n:303

Psoroptes ovis	GCA_002943765.1	93	63,214,126	100.00%	100.00%	86.00%	2,279,290	C:93.7%[S:92.7%,D:1.0%],F:2.0%,M:4.3%,n:303
Sarcoptes scabiei	GCA_000828355.1	19,246	56,251,741	58.80%	41.90%	6.00%	11,383	C:91.1%[S:91.1%,D:0.0%],F:4.0%,M:4.9%,n:303
Steganacarus magnus	GCA_000988885.1	120,241	112,750,608	46.20%	21.60%	0.50%	1,727	C:85.5%[S:84.2%,D:1.3%],F:7.9%,M:6.6%,n:303
Tetranychus urticae	GCF_000239435.1	2,036	89,613,205	99.40%	93.30%	34.60%	212,780	C:93.7%[S:89.4%,D:4.3%],F:1.3%,M:5.0%,n:303
Dermanyssus gallinae	GCA_003439945.1	7,171	959,010,206	100.00%	100.00%	99.80%	278,630	C:95.0%[S:78.2%,D:16.8%],F:0.3%,M:4.7%,n:303
Ixodes ricinus	GCA_000973045.2	205,231	514,471,516	99.80%	99.50%	0.90%	3,060	C:26.4%[S:26.4%,D:0.0%],F:28.1%,M:45.5%,n:303
Ixodes scapularis	GCF_002892825.2	19,746	3,088,623,987	100.00%	100.00%	96.90%	517,316	C:95.4%[S:58.1%,D:37.3%],F:0.3%,M:4.3%,n:303
Ixodes scapularis	GCA_000208615.1	570,637	1,388,472,180	99.90%	98.20%	3.50%	2,942	C:85.1%[S:83.8%,D:1.3%],F:6.6%,M:8.3%,n:303
Metaseiulus occidentalis	GCF_000255335.1	3,993	151,323,873	99.70%	97.80%	34.00%	200,706	C:96.7%[S:93.7%,D:3.0%],F:1.0%,M:2.3%,n:303
Rhipicephalus microplus	GCA_002176555.1	251,890	1,946,541,351	97.40%	94.00%	20.50%	18,585	C:51.9%[S:40.3%,D:11.6%],F:10.2%,M:37.9%,n:303
Tropilaelaps mercedesae	GCA_002081605.1	74,567	326,213,305	64.40%	53.30%	12.80%	13,741	C:90.4%[S:90.1%,D:0.3%],F:5.3%,M:4.3%,n:303
Varroa destructor	GCA_000181155.2	52,152	329,105,442	76.40%	62.10%	21.90%	15,568	C:92.5%[S:90.8%,D:1.7%],F:2.3%,M:5.2%,n:303
Varroa destructor	GCF_002443255.1	4,498	368,670,960	100.00%	99.90%	64.30%	201,886	C:95.7%[S:94.4%,D:1.3%],F:1.7%,M:2.6%,n:303
Varroa jacobsoni	GCF_002532875.1	8,234	365,177,116	100.00%	99.90%	65.10%	96,030	C:96.7%[S:95.4%,D:1.3%],F:1.3%,M:2.0%,n:303
Stegodyphus mimosarum	GCA_000611955.2	159,639	2,694,371,924	72.00%	64.60%	41.40%	46,340	C:88.1%[S:85.1%,D:3.0%],F:5.3%,M:6.6%,n:303

Table 2. KEGG pathway enrichment of 200 most highly expressed genes in fat-feeding mites

Term	Database	ID	Input number	Background number	P-Value	Corrected P-Value
Ribosome	KEGG PATHWAY	dme03010	15	240	2.31E-09	1.04E-07
Spliceosome	KEGG PATHWAY	dme03040	4	128	0.022180427	0.375099591
Protein processing in endoplasmic reticulum						
reticulum	KEGG PATHWAY	dme04141	4	133	0.025006639	0.375099591
Phenylalanine metabolism	KEGG PATHWAY	dme00360	1	8	0.070305572	0.48926869
Arginine and proline metabolism	KEGG PATHWAY	dme00330	2	53	0.072717795	0.48926869
Longevity regulating pathway - multiple species						
multiple species	KEGG PATHWAY	dme04213	2	56	0.079707921	0.48926869
Endocytosis	KEGG PATHWAY	dme04144	3	122	0.080418185	0.48926869
ECM-receptor interaction	KEGG PATHWAY	dme04512	1	12	0.099955241	0.48926869
Fatty acid biosynthesis	KEGG PATHWAY	dme00061	1	13	0.107219727	0.48926869
Oxidative phosphorylation	KEGG PATHWAY	dme00190	3	144	0.115970198	0.48926869

Comment (2) The authors showed that Arachnida species with different diet showed different evolutionary patterns based on some regular analyses. These results are expectable. It is not clear what is the key scientific problem the authors want to solve based on their analyses. What is the purpose of doing these analyses as well as publishing a paper, and how these results will serve the purpose? In the abstract and the conclusions, the authors mentioned twice: “These different genetic bases provide a new perspective for the study of the evolution and diversification of this group, and offers potential drug targets for pest control.” These statements are repetitive, and are very vague. What is the “new perspective” and how to offer potential drug targets? Details should be discussed at least. In addition, not all mite species analyzed in this study are pests. For example, *Metaseiulus occidentalis* is a predatory mite natural enemy.

Thank you for your constructive advice, we have added a description of the motivation of our research in the Abstract and Introduction section to help readers quickly understand the main motivation of our research. Here is a short summary “Diet is one of the most fundamental aspects of an animal’s biology and is a powerful evolutionary force for species adaptation and diversification. Acari (mites and ticks) is one of the most abundant clades of Arachnida, exhibiting extraordinarily diverse dietary types. While studies have focused on morphological and physiological adaptations to different dietary habits, the genetic mechanisms underlying these adaptations are not fully understood. Based on a comparative analysis of 15 Acari genomes and five dietary habits, we found different genetic bases for different diets, mainly related to the need to handle different food types, including increased abilities to find, prepare and digest food.”. Details are in the Introduction section.

Sorry for the confusing expression. We have changed the vague sentence “Based on comparative analyses of 15 Acari genomes, we found genetic bases for three specialized diets. Herbivores experienced stronger selection pressure than other groups; the olfactory genes and gene families involving metabolizing toxins showed strong adaptive signals. Genes and gene families related to anticoagulation, detoxification, and haemoglobin digestion were found to be under strong selection pressure or significantly expanded in the blood-feeding species. Lipid metabolism genes have a faster evolutionary rate and been subjected to greater selection pressures in fat-feeding species; one positively selected site in the fatty-acid amide hydrolases 2 gene was identified. Our research

provides a new perspective for the evolution of Acari and offers potential target loci for novel pesticide development.”. More information has been discussed in the specific dietary section in the revised manuscript.

Our study of evolutionary adaptation for different diets was not only based on pests but based on the mites with specialized dietary styles. However, the genetic adaptation of three specialized diets was implied and could consequently help with inhabitation of some pests such as spider mites, honey bee mites and so on. We could take efficient measures to weak the abilities for finding (olfaction), preparing (detoxification) and digesting (metabolism), which are found in the current study. For example, we have detected positive selection sites in HSP genes involving in olfactory pathways, which offered several drug targets for control the spider mites. Similarly, we have found that the evolutionary speed of lipid metabolism is significantly accelerated in honey bee mites, especially of arachidonic acid lipid metabolism pathway (Figure 6a and 6c). We could design drugs targeting at the candidate genes of arachidonic acid lipid metabolism such as *FAAH2* to inhibit or kill honey bee mites in the following research.

Comment (3) There are many typos and formatting errors in manuscript (including the references). I think the authors should pay more attentions to fix these problems.

Sorry for these mistakes. We have carefully modified our manuscript. And the revised manuscript was edited for proper English language, grammar, punctuation, spelling, and overall style by one or more of the highly qualified native English speaking editors at SNAS (verification code 30F9-7501-1FD4-89B3-2576).

Finally, thank you again for your great suggestions. We hope our manuscript can be of great interest to researchers of related fields now.

Best Regards.

Sincerely,

Dr. De Chen on behalf of all authors

REVIEWERS' COMMENTS:

Reviewer #2 (Remarks to the Author):

Thanks authors' Responses. I have no further comments. I am looking forward to the manuscript published on communication biology.

Reviewer #3 (Remarks to the Author):

I think this manuscript has been well updated, and can be published.